# Psychological Impact during Confinement by COVID-19 on Health Sciences University Students—A Prospective, Longitudinal, and Comparative Study

**DOI:** 10.3390/ijerph19169925

**Published:** 2022-08-11

**Authors:** Luis Iván Mayor-Silva, Manuel Romero-Saldaña, Antonio Gabriel Moreno-Pimentel, Ángela Concepción Álvarez-Melcón, Rafael Molina-Luque, Alfonso Meneses-Monroy

**Affiliations:** 1Faculty of Nursing, Physiotherapy and Podiatry, Complutense University of Madrid, 28040 Madrid, Spain; 2Grupo Asociado de Investigación GA-16, Estilos de Vida, Innovación y Salud, Instituto Maimónides de Investigación Biomédica de Córdoba (IMIBIC), Faculty of Medicine and Nursing, University of Cordoba, 14071 Córdoba, Spain

**Keywords:** coping, COVID-19, nursing and physiotherapy students, personality, resilience, social isolation, stress

## Abstract

Background: The objective was to analyze the factors that influence reactions to confinement situations, such as personality, humor, coping with stressors, and resilience, and to compare this population with a normal situation of exposure to an intense academic stressor such as a partial test, and with the confinement situation caused by the COVID-19 pandemic. Methods: A longitudinal study was performed involving 116 health sciences students from Spain. Three situations were evaluated: a basal situation of normality at the beginning of the course, situation facing an academic stressor (partial test), and confinement situation due to COVID-19. The Positive and Negative Affect Schedule (PANAS), Coping Orientation to Problems Experienced (COPE), Connor–Davidson Resilience Scale, and NEO-FFI scale were used. Results: Significant differences were observed in the increase in negative humor and decrease in positive one, as well as decrease in “Focus on and Venting of Emotions”. Personality factors that better predict humor at confinement were “conscientiousness” for having positive humor and low “extraversion” for negative humor. Conclusions: The confinement situation due to COVID-19 has caused changes in predominant humor, as well as in coping strategies. Personality factors positively or negatively influence the situation.

## 1. Introduction

Before the pandemic, studies showed that health sciences students, especially nursing students, experience significant anxiety and stress throughout their education [1,2]. Some factors make them susceptible to test anxiety [3]. On the other hand, a student who has recently joined a university to study health sciences experiences high anxiety levels [4]. These reactions are related to coping mechanisms, personality factors, and resilience that students put into operation when facing stressful situations [5,6].

Because of the COVID-19 pandemic and the virus’s swift progress, the Spanish government declared a state of alert on 14 March 2020 that had to be extended several times because Spain became a country with a higher death rate per million inhabitants. For this reason, its inhabitants had to remain in a confinement situation for more than 3 months, which was the most restrictive one in Europe, allowing one to go outdoors only for essential activities.

This caused Spanish university students to adapt to a new e-learning methodology, which caused some students to face specific problems in their adaptation [7]. Most nursing students were emotionally affected by high anxiety levels during the COVID-19 outbreak [8].

In confinement, very disparate reactions can be observed in people, which range from sympathetic and creative actions to those who spend hours doing nothing, surrendering to discouragement, and being overwhelmed.

In China, relationships were studied between stressors and the coping style used by university students in the face of SARS, and their psychological adjustment, and it was found that the presence of stressors and an avoidant coping style predicted the presence of symptomatology, although an active type of coping predicted a greater satisfaction with life by controlling such stressors [9]. According to Ucho et al. (2016) [10], personality is an important psychological factor that affects people’s behavior and, therefore, is highly related to the subjects’ reactions and experiences during a confinement situation [10].

At the social level, the context that students had to face was characterized by intense reactions, mainly in the shape of fear of contagion [11], because of the lingering social isolation measures [12,13], loss of loved ones [14], or economic crisis [15]. Lengthier confinement is associated with worse psychological results [12].

In this context, the vision that students could have of their future professional performance was not very hopeful.

For the 2003 SARS pandemic data, Lee et al., (2007) [16] analyzed the psychological consequences in survivors and sanitary and nonsanitary workers a year after the outbreak [16]. The results showed that survivors still had elevated stress levels and worrisome levels of psychological distress. These were higher in employees placed under quarantine who worked in high-risk clinical environments or who had relatives or friends infected with SARS. These results suggested the need to establish specific support and postintervention programs for future health professionals. 

The COVID-19 pandemic has required strong resilience from sanitary professionals and health sciences students. In this context, it was deemed necessary to carry out research to be able to predict and control future sanitary professionals’ reactions and to obtain more resilient profiles that allow confronting present and future pandemics. 

The objectives of this study were as follows:To analyze how personality, humor, coping with stressors, and resilience influence reactions at a confinement situation;To identify which factors can predict better adjustment and what type of person would need greater support in such a situation;To learn the evolution of a cohort of nursing and physiotherapy students in three situations: in a normal situation at the beginning of the academic course, exposed to an intense academic stressor such as a partial test, and during confinement due to the pandemic.

## 2. Materials and Methods

### 2.1. Design, Population, and Sample

An observational and longitudinal study was carried out on the students of the Complutense University of Madrid between October 2019 and May 2020. The target population comprised 370 first-course students from nursing and physiotherapy degrees. Initially, the study was designed to learn coping strategies when facing stress during various times of the 2019–2020 academic year. The abrupt irruption of the COVID-19 pandemic and the subsequent confinement presented a new scenario, a unique opportunity we did not want to squander. After informing them the original (initial) objectives of the study, 116 students agreed to participate (eligible population).

For sample size calculation, the Epidat 4.2 (Department of Sanidade, Xunta de Galicia, Galicia, Spain) tool was used. For an expected resilience mean of 70 points with 10 points standard deviation, 2% precision, and 96% certainty, the sample size obtained was 53 subjects. A randomized sample size was made, stratified by age and sex, to configure the initial study sample (Situation 1—basal state without stress) of 78 participants (Figure 1).

### 2.2. Evaluation States or Situations

Three states or situations were evaluated where the students faced different stressing contexts:

Situation 1: basal state without stress. This was the initial context of the study in October 2019. The students had begun their first course studies in nursing or physiotherapy degrees. This represents a new context marked by uncertainty that presumes the challenge of starting university studies: everything is new and unsettling—new classmates, professors, subjects, work and study methods, roles, etc. 

Situation 2: state from academic stressor. This was the state prior to tests. This corresponds to the period of the first partial tests for degree students in January and February 2020. Students deal with the challenge of facing the first evaluation of their knowledge in their first course at university.

Situation 3: state of confinement due to pandemic. On the third week of March 2020, due to the COVID-19 pandemic, the university’s onsite lectures were discontinued all over Spain, and home confinement was established to guarantee social distancing among the population, which generated a unique situation experienced by the students who had to adapt to this new context: virtual lectures, adaptation of theoretical–practical contents, the use of new information and communication technologies, abrupt change in learning habits and those of relating with the university community, etc.

### 2.3. Measurement Scales and Variables

The following sociodemographic variables were recorded: sex (men or women), age (years), degree (nursing or physical therapy), and employment status (working or studying).

On each of the three situations previously described, the participants’ psychological factors were analyzed, and the coping strategies implemented during the exposure to each context, as well as the students’ resilience, were evaluated. The following validated self-administered questionnaires were employed:

Positive and Negative Affect Schedule (PANAS) separately assesses the positive and negative emotional experiences one has recently lived [17]. This is a 20-item questionnaire where participants respond using Likert scales. The items are organized into two groups: 10 items refer to positive aspects and 10 to negative aspects. Likert scales range from 1 to 5, where 1 = slightly or almost nothing and 5 = extremely. Scale scores range from a minimum of 10 to a maximum of 50, with no categories. 

Coping Orientation to Problems Experienced (COPE). The Spanish version of the COPE-48 scale [18] assesses the following 8 coping strategies using a Likert scale ranging from 1 = “I never do that” to 4 = “I do that frequently”: Active Problem-Focused Coping, Alcohol/Drug Disengagement, Focus on and Venting of Emotions, Seeking Social Support, Humor, Turning to Religion, Denial, and Restraint Coping.

Connor–Davidson Resilience Scale. The Spanish adaptation consists of 25 items that participants evaluate using Likert scales ranging from 0 (not at all) to 4 (almost always). The items are grouped into 5 dimensions: Persistence–Tenacity–Self-Efficacy, Control Under Pressure, Adaptability and Support Networks, Control and Meaning, and Spirituality. The sum of these values constitutes the total value for resilience, whose thresholds are as follows: less than 70 (low), 70 to 87 (intermediate), and greater than 88 (high).

NEO-FFI scale. The NEO Five-Factor Inventory is a questionnaire consisting of five factors designed to assess personality in different contexts [19]. It is made up of 60 sentences, which are evaluated using a Likert scale that ranges from 0 (not at all) to 4 (almost always). The five dimensions assessed are: Neuroticism, Extraversion, Openness, Agreeableness, and Conscientiousness. 

### 2.4. Ethical and Legal Aspects

The principles enshrined in the Declaration of Helsinki on Biomedical Research Involving Human Subjects were observed at all times. All the students were informed of the objectives and terms of implementation of the research and signed an informed consent form that explained that participation was completely voluntary and anonymous, they could freely withdraw from the study at any time without giving reason, and such participation did not entail any benefit or harm for them. The confidentiality and privacy of their information were observed in compliance with current regulations on the protection of personal data. The data were entered in secure databases, and access to them was restricted to researchers only. Data analysis was limited to the purposes of this study. The research protocol was approved by the faculty’s Research Committee.

### 2.5. Statistical Analysis

Means and standard deviations were calculated for quantitative variables. Absolute and relative frequencies (percentages) were calculated for qualitative variables. The assumption of normality of data was checked using graphical representation tests (histograms and Q–Q and P–P plots) and statistical significance tests such as the Shapiro–Wilk or Kolmogorov–Smirnov test (with the Lilliefors correction applied).

For statistical comparisons, the *χ*-square test was used (applying Fisher’s exact test, when indicated), and the *Z*-test was used for qualitative variables. For the comparison of two means, Student’s *t*-test or Mann–Whitney’s *U*-test was used, depending on whether or not the data in question were parametric, respectively. For the comparison of three arithmetic means, the parametric test ANOVA was used, with post hoc analyses for the pairwise comparison of groups, using the Bonferroni, Tukey’s HSD, and Scheffe tests. The homoscedasticity of the data was checked using Levene’s test. The comparison of the three paired means was carried out using the ANOVA test for paired means.

All hypothesis tests were two-tailed and conducted with a statistical significance threshold of alpha error <5% (*p* < 0.05). Confidence intervals were calculated with 95% certainty. SPSS v. 22 (SPSS, IBM, Chicago, IL, USA) and Epidat v. 4.2 ((Department of Sanidade, Xunta de Galicia, Galicia, Spain) were used for data analysis.

## 3. Results

Out of the 370 enrolled students in the first course of nursing and physiotherapy degrees, 293 were women (79.2%) and 77 were men (20.8%). A total of 116 agreed to participate in the study. From those, a random sample of 78 participants was selected who started the study in Situation 1 (basal state) in October 2019: 62 women (79.5%) and 16 men (20.5%). During the 8-month follow-up, 28 participants dropped from the study: 9 of them prior to the tests (Situation 2) on February 2019 and 19 prior to confinement (Situation 3). A total of 50 students completed the study, comprising the sample for the comparative analysis of the three evaluated situations (Figure 1). 

Out of the 50 students, 49 were women (98%) and 46 were enrolled in nursing (92%). The age average was 19.9 (5.1) years with a 95% CI of 18.5–21.4, and the range was between 17 ears (minimum) and 41 years (maximum). Thirty-two percent of the participants were working while studying. Table 1 gathers the study sample’s characteristics and those of the participants who dropped during the 8-month follow-up. 

The personality characteristics of the participants in the basal situation were evaluated using the NOE-FFI questionnaire (Table 2). The participants showed a very high level of neuroticism: 86% of the sample was above the 50th percentile of adult population; the Agreeableness and Conscientiousness dimensions were clearly under the adult population mean (30%). 

Table 3 shows the results from the inventories for the undertaken positive and negative affect (PANAS), coping strategies (COPE-48), and resilience (Connor–Davidson) scales for the three situations experienced by the students throughout the study.

Regarding the PANAS scale, significant changes were observed during the confinement period (State 3) both for positive and negative affect. While positive affect decreased (*p* < 0.001), the perceived negative affect increased (*p* < 0.01) with respect to the basal situation (State 1) and the pretest situation (State 2). Regarding the coping strategies for stress set into motion by students, the only one that showed a significant variation among the three evaluated situations was “Focus on and Venting of Emotions”. Its use significantly decreased between States 1 and 3 (*p* < 0.05). Finally, the Resilience scale did not present any significant variation among the evaluated situations neither for its dimensions nor for the total score, which ranged between 68.7 and 69.2 points.

Lastly, the influence of the participants’ personality in the basal situation (State 1) was studied concerning the experienced affect and resilience during the confinement and social distancing stage caused by the COVID-19 pandemic (State 3) (Table 4).

From the results presented in Table 4, resilience was higher during confinement in those subjects who basally showed less neuroticism compared with students with basal neuroticism higher than the 50th percentile (78.9 vs. 67.6, *p* < 0.05). Furthermore, the suffered negative affect increased in the group of students with higher basal extraversion (28.3) compared with students with lower basal extraversion (28.3 vs. 24.5, *p* = 0.08). The experienced positive affect was higher on those students who, in the basal state, presented a greater degree of conscientiousness (26.3) compared with participants with basal conscientiousness lower than the 50th percentile (26.3 vs. 21.7, *p* < 0.05). 

## 4. Discussion

Scores on positive and negative affect changed during the confinement situation compared with the other two analyzed situations. We ascertained how positive affect significantly decreased and how the negative affect increased even more than facing the academic stressor that implicitly carried the novelty of being the first of the degree’s tests. These data match those found in similar populations in Chile or China during the confinement caused by COVID-19 [20], which makes us think that the confinement situation had very different characteristics compared with other types of stressors such as the anticipatory anxiety when facing tests and perhaps requires coping strategies different from those when preparing for a test. Furthermore, we have to take into account the sample characteristics. Youth seems to represent a risk factor for poor functioning during the pandemic [21]. These results could be perceived as counterintuitive because symptoms and consequences of the new coronavirus are worse for the elderly than young adults according to one study [22], and the youngest population can use computer resources to study, relate, and have fun with greater ease than the oldest population.

With regard to the coping variable, significant data were only obtained in the decrease in the “Focus on and Venting of Emotions” strategy during confinement compared with the other analyzed situations. The results may explain that the subjects stopped using this coping style because it was not as effective during confinement, because this strategy may affect coexistence with others. These scores can be compared with the studies on the Chinese population where the relationship between stressors and the coping style used by university students facing SARS, and their psychological adjustment, was evaluated, showing that the presence of stressors and an avoidant coping style predicted the presence of psychopathological symptomatology, while an active coping style predicted a greater satisfaction with life by controlling such stressors [9]. Perhaps a way to maintain good mental health at home is to minimize this avoidant coping mechanism, particularly in a situation where its length is not well-defined.

In regard to resilience, statistically significant results were not observed in our study for the three analyzed situations. There are studies that showed that it is not a variable with a predictive value; it is very feeble and decreases even more once personal recovery ability is taken into account [23]. Furthermore, another study showed a clear relationship between resilience and satisfaction in a confinement situation [24]. Perhaps this was due to our sample’s particular characteristics, consisting of health sciences students experiencing higher pressure to aid other people.

Regarding the analyzed personality variables, the results indicated that subjects who showed higher neuroticism improved in positive affect during confinement, and those with higher scores in extraversion showed increased negative affect scores.

Our aim was to determine if there is a personality that may be more resilient to the confinement situation compared with other stressing situations. The results showed an interesting relationship between greater resilience in subjects with lesser basal neuroticism; this finding corresponds with those of the majority of psychological studies that examined both variables in university populations. Some authors concluded that resilience increases inasmuch as the subject presents personality characteristics related to emotional stability, conscientiousness, and extraversion [25,26].

It is to be noted that the resilience scores were higher, but the positive affect during confinement was not necessarily greater in the more resilient subjects. Instead, a greater positive affect was observed in subjects with higher basal conscientiousness. These data are similar to those obtained in a Bolivian population during a confinement situation, where a greater negative affect with lesser basal extraversion stood out; this demonstrated that the lack of extraversion, even in a situation where people are not able to communicate in-person due to confinement, impacts the state of mind [27].

Among the study limitations, having a voluntary sample may have influenced some of the variables we were studying. Future studies must include different types of samples to research other forms of psychological impact, or even within the student population; it would have been very useful to have carried out this research with students who volunteered or worked at hospitals. However, having data from the same population in three different situations is a strength of this study. More studies that perform follow-ups on the population are needed, given that the “new normality” is anticipated to be marked by intermittent confinements where students will have to continue adapting to this circumstance to become the professionals of tomorrow.

## 5. Conclusions

Taking into account the obtained results, the following conclusions on the factors that influence confinement were reached:The confinement situation presented stress characteristics different from those of an academic stress situation.A distinctive factor of the COVID-19 confinement was changes in affect, increasing negative and decreasing positive affect.The coping strategy that decreased during the confinement situation, perhaps due to its inefficacy, was “focus on and venting of emotions”.There as greater resilience during the confinement situation in subjects with lesser basal neuroticism.Subjects who presented conscientiousness as a personality factor further developed positive humor.Negative humor during the pandemic was particularly shown by subjects with lesser basal extraversion.

## Figures and Tables

**Figure 1 ijerph-19-09925-f001:**
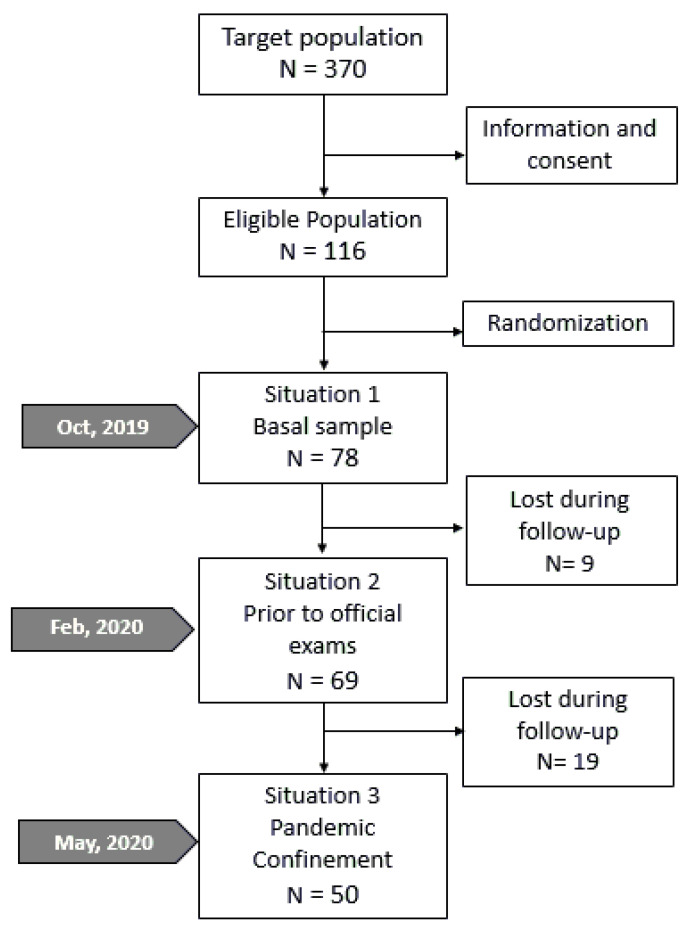
Study follow-up flowchart. Situation 1: basal state; situation 2: pretesting; and situation 3: confinement due to pandemic.

**Table 1 ijerph-19-09925-t001:** Characteristics of the study sample.

Variable	Sample *n* = 50	Drops during the Study *n* = 28	*p*
	Mean or *n* (SD or %)	Mean or *n* (SD or %)	
**Men**	1 (2)	15 (53.6)	<0.001
**Women**	49 (98)	13 (46.4)	
**Age (years)**	19.9 (5.1)	19.8 (5.7)	NSS
**Employed**	16 (32)	8 (28.6)	NSS
**Nursing**	46 (92)	18 (64.3)	<0.01
**Physiotherapy**	4 (8)	10 (35.7)	

SD: standard deviation; NSS: nonstatistically significant.

**Table 2 ijerph-19-09925-t002:** Sample’s personality dimensions according to the NEO-FFI scale.

Dimension	Mean (SD)	*p* (50) Adult Population	Score ≥ *p* (50) *n* (%)
**Neuroticism**	25.3 (9.1)	16	43 (86)
**Extraversion**	32.3 (8.1)	34	24 (48)
**Openness**	32.1 (6.7)	30	30 (60)
**Agreeableness**	30 (6.7)	34	15 (30)
**Conscientiousness**	32.4 (6.9)	36	15 (30)

SD: standard deviation; *p* (50): 50th percentile.

**Table 3 ijerph-19-09925-t003:** Results of the PANAS, COPE-48, and Connor–Davidson Resilience scales according to study situations.

	State 1 *n* = 50	State 2 *n* = 50	State 3 *n* = 50	*p* *
	Mean (SD)	Mean (SD)	Mean (SD)	
**PANAS Scale**				
**Positive affect**	32.8 (6.3)	33.4 (5.8)	24.9 (7.7)	<0.001
**Negative affect**	22.1 (7.4)	21.6 (7.2)	26.3 (7.7)	<0.01
**COPE-48 Scale**				
**APFC**	25.9 (5)	25.8 (3.8)	25.3 (3.8)	NSS
**ADD**	4.6 (1.4)	4.7 (1.4)	4.6 (1.6)	NSS
**FVE**	11.4 (3)	11 (3)	10.5 (2.6)	<0.05
**SSS**	23 (5.3)	23.7 (5.7)	22.3 (5.7)	NSS
**HUM**	9.4 (3.6)	9.2 (3.5)	9.3 (3.3)	NSS
**TTR**	6.9 (4.3)	7.1 (4.4)	7.2 (4.6)	NSS
**DEN**	5.2 (2)	5.5 (2.1)	5.3 (2.1)	NSS
**RC**	30.5 (5.7)	31.5 (5.2)	30.8 (4.7)	NSS
**Resilience Scale**				
**PTS**	23.7 (4.5)	23.8 (4.5)	23 (4.9)	NSS
**CUP**	16.8 (4.4)	17.2 (3.9)	17 (4.4)	NSS
**ASN**	14.9 (3.2)	15.6 (3.1)	16.1 (3.5)	NSS
**CAM**	8.7 (2.1)	8.9 (2)	8.7 (2.3)	NSS
**SPR**	4.6 (2.4)	4.3 (2.3)	4.3 (2.5)	NSS
**RSC**	68.7 (11.4)	69.8 (11)	69.2 (13.7)	NSS

APFC: Active Problem-Focused Coping; ADD: Alcohol/Drug Disengagement; FVE: Focus on and Venting of Emotions; SSS: Seeking Social Support; HUM: Humor; TTR: Turning to Religion; DEN: Denial; RC: Restraint Coping; PTS: Persistence, Tenacity, Self-Efficacy; CUP: Control under Pressure; ASN: Adaptability and Support Networks; CAM: Control and Meaning; SPR: Spirituality; RSC: Resilience. State 1: Basal. State 2: Two days before testing. State 3: Confinement. * ANOVA test for comparison of three repeated means; SD: standard deviation; NSS: nonstatistically significant.

**Table 4 ijerph-19-09925-t004:** Personality profiles according to NEO-FFI in States 1 and 3 result (Confinement).

	**Neuroticism**			**Extraversion**			**Openness**		
** Confinement (state 3) **	<*p* (50) *n* = 7	≥*p* (50) *n* = 43	*p* *	<*p* (50) *n* = 26	≥*p* (50) *n* = 24	*p* *	<*p* (50) *n* = 20	≥*p* (50) *n* = 30	*p* *
** Positive affect **	20.6 (5.5)	25.6 (7.8)	0.1	26.2 (8.1)	23.5 (7.7)	0.23	24.3 (8.3)	25.3 (7.4)	0.7
** Negative affect **	30.5 (6.3)	26.3 (7.7)	0.1	24.5 (7.3)	28.3 (7)	0.08	26 (7)	26.5 (8.11)	0.84
** Resilience **	78.9 (11.2)	67.6 (13.6)	<0.05	67.2 (14.1)	71.4 (13.4)	0.29	69.2 (16.2)	69.2 (12.4)	0.99
	**Agreeableness**			**Conscientiousness**					
** Confinement (state 3) **	<*p* (50) *n* = 15	≥*p* (50) *n* = 35	*p* *	<*p* (50) *n* = 15	≥*p* (50) *n* = 35	*p* *			
** Positive affect **	25.3 (8)	23.5 (7)	0.49	26.3 (7.5)	21.7 (7.4)	<0.05			
** Negative affect **	25.9 (7.7)	27.8 (7.7)	0.47	25.4 (7.9)	28.4 (6.8)	0.21			
** Resilience **	68.7 (14)	70.9 (12.8)	0.63	68.7 (12.8)	70.4 (16.1)	0.69			

*p* (50): 50th percentile. * Student’s *t*-test/Mann–Whitney *U*-test for comparing two independent means. All measures are shown as mean (SD). SD: standard deviation.

## Data Availability

Not applicable.

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
