# Peer review of "Psychological Impact during Confinement by COVID-19 on Health Sciences University Students—A Prospective, Longitudinal, and Comparative Study"

_ijerph, 2022, doi:10.3390/ijerph19169925_

Round 1

Reviewer 1 Report

In general, the article complies with the formal and methodological requirements of the journal. The objective explicitly expressed in the abstract of analyzing the factors that influence reactions to the lock-in situation, such as personality, mood, stress coping and resilience, and comparing this population in normal situation, exposed to intense academic stress as a partial test, and to the lock-in situation provoked by the COVID 19 pandemic, is met.

The three working hypotheses (lines 75-81) have been duly verified through an observational and longitudinal study using previously verified instruments. The COPE (Coping Orientation to Problems Experienced); the Connor-Davidson resilience scale; the NEO-FFI scale.

The graphs and tables are correct and the results clearly show the results obtained. Likewise, in the discussion the results obtained are summarized and extensively argued, citing the sources and previous studies that support the authors' thesis. The conclusions are correct and the bibliography is sufficiently updated.

Although it would have been interesting to expand the sample, including more universities or more degrees to make a comparison and that: "Among the limitations of the study is the fact of having a voluntary sample, which may have influenced some of the variables we are studying" (line 345) or that "within the student population it would have been very useful to have conducted this research with students who were volunteers or worked in hospitals" (line 347-348) as the authors comment. I consider that the manuscript can be published without further modification.

Author Response

Dear Reviewer,

Thank you very much for your comments.

Sincerely,

The authors

Reviewer 2 Report

The arcicle addresses important aspects of the lives of students in COVID-19 pandemic. 

Unfortunately in my opinion, it cannot be published in IJERPH in this form.

The study group is too small for this type of research. 

The DISCUSSION is not insective. There are only 8 references in it...!? 

The Discussion should be more carefully conducted.

REFERENCES count only 20 items, this is too small number.

I suggest that the authors conduct research on a largen group of students. Select the literature for the discussion more carefully.

Author Response

Dear reviewer:

Thank you very much for your comments.

Sincerely,

The authors

Reviewer 3 Report

Next, a series of comments and suggestions are presented so that, if the authors of the manuscript believe so, they can improve some aspects related to the work presented.

1. Introduction

The introduction should have an initial part that responds to the primary objective of the research, which was to compare various variables in a group of students at the beginning of the course and at the beginning of the exam period. Nothing is said about this in the introduction. It only focuses on everything related to COVID and confinement, losing sight of that first objective, which was the one that actually launched this study. We cannot ignore everything that concerns the change in the personality of the participants from the beginning of the course to the first exams, since it is concluded that one does not act in the same way before an exam and before confinement.

In this sense, the bibliography cited is scarce in studies that analyze these first data, most of them entering studies related to the pandemic.

Likewise, it would be necessary to expand with a multitude of studies on these analysis variables that have been carried out at the national level, since a multitude of works emerged in this regard.

2. Material and method

Here is rather an assessment, a thought out loud. Having only 50 students for the study is a shame. I believe that researchers should make an effort to make our students understand that they should participate in this type of study. As future health professionals, they should be more involved in research aimed at improving people, not only from a clinical perspective, but also from a social perspective.

In reference to the sex of the participants, an annotation. It should be allowed, if it has not been the case, that students could have marked more options instead of only male or female. It is important that from a scientific field and from a health field, we begin to normalize these issues and not leave aside those other subjects who may not be represented with this duality.

Regarding the ethical aspects, although it is specified that the participants were informed about the objectives and execution conditions, it is not indicated whether or not they had the authorization of the ethics committee of the universities from which the authors come.

3. Results [in relation to population and sample]

I do not fully understand that, if a random sample was carried out, stratified by age and sex, to configure the initial sample of the study, at the end, in the analysis section there were 49 women and only one man. I do not know, because such data has not been reported, how the 370 subjects that make up the study population were distributed according to sex. In addition, if one aspect that differentiated them was the degree, this characteristic should have been made up of a stratum, not that in the end there are 46 nursing students and only 4 physiotherapy students. It would almost have been better to talk about health degree students instead of specifying, since the very low number of physiotherapy participants is not going to help us to see if the profile varies with respect to that of nursing students. The same occurs with the gender variable.

Nothing has been taken into account of the employment situation variable throughout the investigation. It is better not to include it. The same occurs with the age variable, a variable that has only been used to establish the sample, but we do not know the age range of the participants and their mean, nor is there any analysis or conclusion in which this variable is collected.

Reviewing table 1, the only two variables that have significant differences are sex and degree, but we have already mentioned the difference in subjects (Sex: Male=1/Female=49; Nursing=46/Physiotherapy=4).

Based on these data, it would almost have been more desirable to perform the analyzes without distinguishing between these variables.

The really interesting information is that which appears in table 3 in which a comparison is made with the three situations.

4. Discussion

It would have been desirable to compare the results obtained on academic stress with other studies. In the end, everything seems more oriented towards COVID and it seems, according to the authors, that the results do not differ between situations of academic stress and situations of confinement. Although the two questions are important, knowing everything related to academic stress must be a priority in order to take some type of action. It's not that it's not necessary with COVID, but academic stress, by nature, should show up more in students' academic lives than being locked down again by a COVID-like virus.

5. Finally,

If citations are being used with the Vancouver standard, I don't understand that throughout the text authors and years appear at certain times, this is more typical of other types of citation standards.

Author Response

REVIEWER 3

Responses to reviewer comments # 3

Next, a series of comments and suggestions are presented so that, if the authors of the manuscript believe so, they can improve some aspects related to the work presented.

Response: Thank you very much

  1. Introduction.

1.1. Reviewer: The introduction should have an initial part that responds to the primary objective of the research, which was to compare various variables in a group of students at the beginning of the course and at the beginning of the exam period. Nothing is said about this in the introduction. It only focuses on everything related to COVID and confinement, losing sight of that first objective, which was the one that actually launched this study. We cannot ignore everything that concerns the change in the personality of the participants from the beginning of the course to the first exams, since it is concluded that one does not act in the same way before an exam and before confinement.

In this sense, the bibliography cited is scarce in studies that analyze these first data, most of them entering studies related to the pandemic.

Response: The authors agree with this comment. The Introduction section has been rewritten, adding two new paragraphs and new references about the primary objective of the study.

New text added in the manuscript:

 First paragraph: Before the pandemic, studies show that health sciences students, especially nursing students, present significant problems of anxiety and stress throughout their education [1,2]. Some factors make them susceptible to test anxiety [3]. On the other hand, a student who has recently joined the university to study health science generates high anxiety levels [4]. These reactions are related to coping mechanisms, personality factors and resilience that students put into operation when facing stressful situations [5,6]”.

Second paragraph: This caused Spanish university students to adapt to a new e-learning methodology, which caused some students to develop specific problems in their adaptation [7]. Most nursing students were emotionally affected by high anxiety levels during the COVID-19 outbreak [8]”.

1.2. Reviewer: Likewise, it would be necessary to expand with a multitude of studies on these analysis variables that have been carried out at the national level, since a multitude of works emerged in this regard.

Response: The following paragraph has been added in the manuscript.

Third paragraph: At a social level, the context that the students had to face is characterized by intense reactions, mainly in the shape of fear of contagion [11], because of the lingering social isolation measures [12,13], the loss of loved ones [14] or due to the economic crisis [15]. Lengthier confinement is associated with worse psychological results [12].

In this context, the vision the students could have of their future professional performance was not very hopeful”.

  1. Material and method

2.1. Reviewer: Here is rather an assessment, a thought out loud. Having only 50 students for the study is a shame. I believe that researchers should make an effort to make our students understand that they should participate in this type of study. As future health professionals, they should be more involved in research aimed at improving people, not only from a clinical perspective, but also from a social perspective.

Response: The authors agree with this idea, and therefore we inform students about the importance of research to clarify research questions and, from this point of view, we try to motivate students to participate in research studies that are done in University. However, sometimes we do not fully achieve the objective set.

2.2. Reviewer: In reference to the sex of the participants, an annotation. It should be allowed, if it has not been the case, that students could have marked more options instead of only male or female. It is important that from a scientific field and from a health field, we begin to normalize these issues and not leave aside those other subjects who may not be represented with this duality.

Response: Thank you very much for this comment. The authors agree with this “annotation”. The gender perspective must include in all aspects of the study design, and the sex variable must be a polycotomic variable not only male or female.

2.3. Reviewer: Regarding the ethical aspects, although it is specified that the participants were informed about the objectives and execution conditions, it is not indicated whether or not they had the authorization of the ethics committee of the universities from which the authors come.

Response: In the ethical and legal aspects subsection, we had indicated, "The research protocol was approved by the faculty’s Research Committee". We have the original document available to the editor and reviewers.

  1. Results [in relation to population and sample]

3.1. Reviewer: I do not fully understand that, if a random sample was carried out, stratified by age and sex, to configure the initial sample of the study, at the end, in the analysis section there were 49 women and only one man. I do not know, because such data has not been reported, how the 370 subjects that make up the study population were distributed according to sex.

Response: The subsection “Design. Population. Sample” indicates that the target population was made up of 370 first-year students corresponding with Degrees in Nursing and Physiotherapy. The calculation of the sample size obtained a sample of 78 randomly selected, stratified by age and sex. Throughout the 8 months of the study, 28 participants were lost (Figure 1).

In the Results, the distribution of the gender variable in the study population (n=370) and in the initial sample (n=78) has been specified.

On the other hand, Table 1 shows the distribution of the initial sample (78 students), of which 16 were men (20.5%) and 62 women (79.5%).New text add in manuscript (in red):

New text added in the manuscript (in red)

Out of the 370 enrolled students in the first course of Nursing and Physiotherapy Degrees, 293 were females (79.2%) and 77 males (20.8%). A total of 116 agreed to participate in the study. From those, a random sample of 78 participants was selected who started the study in Situation 1 (basal state) in October, 2019 62 females (79.5%) y 16 males (20.5%).

3.2. Reviewer. In addition, if one aspect that differentiated them was the degree, this characteristic should have been made up of a stratum, not that in the end there are 46 nursing students and only 4 physiotherapy students. It would almost have been better to talk about health degree students instead of specifying, since the very low number of physiotherapy participants is not going to help us to see if the profile varies with respect to that of nursing students. The same occurs with the gender variable.

Response: Thank you very much for your comment. The sampling was carried out stratifying by age and sex, and the Degree was not taken into account. In the initial sample of 78 students, there were 14 Physiotherapy Degree students, but 10 dropped out of the study. The authors have considered it appropriate to indicate this circumstance in Table 1. Furthermore, the study title shows “health sciences university students” without differentiating between Degree.

 3.3. Reviewer: Nothing has been taken into account of the employment situation variable throughout the investigation. It is better not to include it. The same occurs with the age variable, a variable that has only been used to establish the sample, but we do not know the age range of the participants and their mean, nor is there any analysis or conclusion in which this variable is collected.

Response: Thank you very much for this comment. The variable "employment" has been used only to describe the sample, therefore, it was included in Table 1. About the age variable, the mean age, standard deviation and 95% confidence interval were collected in the manuscript. Now the age range has been added.

New text add in manuscript (in red):

Out of the 50 students, 49 were female (98%) and 46 pertained to Nursing (92%). Age average was 19.9 (5.1) CI 95% (18.5-21.4) and the range was 17  (minimum) and 41 (maximum).

3.3. Reviewer: Reviewing table 1, the only two variables that have significant differences are sex and degree, but we have already mentioned the difference in subjects (Sex: Male=1/Female=49; Nursing=46/Physiotherapy=4). Based on these data, it would almost have been more desirable to perform the analyzes without distinguishing between these variables.

The really interesting information is that which appears in table 3 in which a comparison is made with the three situations.

Response: Thank you very much for your comment. The authors fully agree with the comment. Table 1 only shows the sociodemographic variables that characterize the sample, while the most valuable results are shown in Table 3 and Table 4, where the impact of the three stress situations on personality variables, stress coping strategies, resilience, etc.

  1. Discussion

4.1. Reviewer: It would have been desirable to compare the results obtained on academic stress with other studies. In the end, everything seems more oriented towards COVID and it seems, according to the authors, that the results do not differ between situations of academic stress and situations of confinement. Although the two questions are important, knowing everything related to academic stress must be a priority in order to take some type of action. It's not that it's not necessary with COVID, but academic stress, by nature, should show up more in students' academic lives than being locked down again by a COVID-like virus.

Response: Thank you very much for this comment.

  1. Finally,

5.1. Reviewer: If citations are being used with the Vancouver standard, I don't understand that throughout the text authors and years appear at certain times, this is more typical of other types of citation standards.

Response: Thank you very much for this comment. The authors have corrected these citations.

Round 2

Reviewer 2 Report

Non